# Pest Management Challenges and Control Practices in Codling Moth: A Review

**DOI:** 10.3390/insects11010038

**Published:** 2020-01-03

**Authors:** Martina Kadoić Balaško, Renata Bažok, Katarina M. Mikac, Darija Lemic, Ivana Pajač Živković

**Affiliations:** 1Department for Agricultural Zoology, Faculty of Agriculture, University of Zagreb, Svetošimunska 25, Zagreb 10000, Croatia; rbazok@agr.hr (R.B.); dlemic@agr.hr (D.L.); ipajac@agr.hr (I.P.Ž.); 2Centre for Sustainable Ecosystem Solutions, School of Earth, Atmospheric and Life Sciences, Faculty of Science, Medicine and Health, University of Wollongong, Wollongong 2522, Australia; kmikac@uow.edu.au

**Keywords:** codling moth, resistance mechanisms, genetics, control strategies, anti-resistance program, geometric morphometrics, SNPs

## Abstract

The codling moth, *Cydia pomonella* L., is a serious insect pest in pome fruit production worldwide with a preference for apple. The pest is known for having developed resistance to several chemical groups of insecticides, making its control difficult. The control and management of the codling moth is often hindered by a lack of understanding about its biology and ecology, including aspects of its population genetics. This review summarizes the information about the origin and biology of the codling moth, describes the mechanisms of resistance in this pest, and provides an overview of current research of resistant pest populations and genetic research both in Europe and globally. The main focus of this review is on non-pesticide control measures and anti-resistance strategies which help to reduce the number of chemical pesticides used and their residues on food and the local environment. Regular monitoring for insecticide resistance is essential for proactive management to mitigate potential insecticide resistance. Here we describe techniques for the detection of resistant variants and possibilities for monitoring resistance populations. Also, we present our present work on developing new methods to maintain effective control using appropriate integrated resistance management (IRM) strategies for this economically important perennial pest.

## 1. Introduction

### Origin and Biology of the Codling Moth, Cydia pomonella

The codling moth (CM) (*Cydia pomonella* L.) is a key pest in most pome fruit orchards in Croatia and worldwide. This pest, besides apple, also is a pest of pear, walnut, quince and some stone fruits where it causes economic losses in fruit production [1]. Balachowsky and Mesnil [2] were the first to mention CM, and provided data on its origin and damages caused to fruit historically. In Croatia, according to Kovačević [3], CM has been present since ancient times. In North America, it is known that the pest was introduced ca. 1750 [4]. CM was originally from Eurasia, most likely Kazakhstan, but interestingly it was not reported in China until 1953 [5]. Over the last two centuries it dispersed globally with the cultivation of apples and pears. Currently, CM is present in South America, South Africa, Australia and New Zealand [6]. CM occurs in almost every country where apples are grown, and it has achieved a nearly cosmopolitan distribution, being one of the most successful pest insect species known today [7].

CM adults are small (~10 mm in length). They can be distinguished from other moths associated with fruit trees by their dark brown wingtips that have shiny, coppery markings [8]. It overwinters as a fully grown larva within a thick, silken cocoon that can be found under loose scales of bark and in the soil or debris around tree bases [9]. The larvae pupate inside their cocoons in early spring when temperatures exceed 10 °C. Depending on ambient temperature, pupal development occurs within 7–30 days. For the development of adults, the sum of 100 degree-days measured from the 1st of January are required [10]; this value is usually attained at the end of April (i.e., northern hemisphere growing season). For one whole generation of CM, the sum of 610 degrees is required for the complete development of the insect, i.e., from eggs until the appearance of adult moths [10]. A second generation appears after ten days and its flight and egg laying lasts from mid-July to mid-August. Diapausing larvae overwinter in their hibernacula, pupate and then emerge the following spring [11].

The CM has adapted successfully to different habitats by forming various ecotypes, often designated by the term ‘strains’, which differ among each other in several morphological, developmental and physiological features [12]. On apples and pears, larvae penetrate fruit and bore into the core, leaving brown-colored holes in the fruit that are filled with frass (larval droppings) [8]. If chemical treatment is not used during production, CM can cause a decrease in apple harvest from 30% up to 50%. For apples, intensive production tolerates 1% of infested fruit. Producers, with various methods of fruit protection, try to lower that number below 0.5% [1,3].

Depending on the cultivation area and climatic conditions, the pest develops one to four generations/year. According to Neven [13,14], CM diapause can be facultative and depends on both photoperiod and temperatures. The overwintering generation emerges synchronously in the spring followed by one to two slightly overlapping emergence peaks later on in the season. The CM life cycle can be affected by temperature and day length, resulting in different emergence patterns. Pajač et al. [15] confirmed that there is a possibility that an additional (third) generation of the pest can develop in Croatia in years in which the sum of degree-days is higher than the average. CM abundance cannot be explained by any single ecological factor [16]. Following the dynamics and abundance of CM adults over a 10-year period (2000–2009) Pajač and Barić [17] observed marked differences in their population dynamics. Their research confirmed the earlier appearance of adults in the early season and associated longer flight times. Also, the total number of adults caught in pheromone traps increased as the maximum daily number of moths caught per trap also increased. As the climate has changed and higher daily and annual temperatures are recorded, it is thought that this has a resulting impact on the biology of this pest. It is this global phenomenon coupled with chemical-resistant CM biotypes that could be responsible for the longer flight period and observed overall increase in abundance of CM.

## 2. Insecticides Resistance

In apple orchards, 70% of insecticides used are to control CM [6]. CM control is achieved using various neuroactive products such as organophosphates, carbamates, synthetic pyrethroids, neonicotinoids, and insect growth regulators (IGR). The CM is a very plastic species and easily adapts to different climatic conditions including the development of resistance to various groups of synthetic insecticides in the USA and Europe [6,18,19,20]. According to May and Dobson [21], the spread of resistance in insect populations depends on multiple factors, including: the intensity of insecticide selection pressure, the migration ability of individuals, and the fitness costs linked with resistance. In the CM, the first case of resistance recorded was to arsenates in 1928 in the USA [22]. Since then, new cases of resistance have been reported in almost all of the main apple-growing regions worldwide [18,23,24,25]. During the 1980s and 1990s CM control in Europe was achieved using broad spectrum insecticides (pyrethroids and organophosphates [OP]), however, the evolution of pesticide resistance efficacy for these insecticides diminished quickly [18,20,26,27]. Reyes et al. [28] states that insecticide resistance in CM in Europe was first detected ca. 1990 to diflubenzuron (in Italy and southeastern France); further pesticide control failures were observed in Switzerland and Spain. CM populations are now resistant to neonicotinoids including environmentally friendly avermectins [28]. Further, CM has developed resistance to azinphos-methyl and tebufenozide in post-diapausing larval stages, to OP [29] insecticides and more recently to insect growth regulators (IGRs). Resistance is mainly associated with the detoxification system’s mixed-function oxidases (MFO), glutathione-S-transferases (GST) and esterases (EST) [18,28,30]. A kdr mutation in the voltage-dependent sodium channel is involved in resistance to pyrethroids [31] and an acetylcholinesterase (AChE) mutation has been identified in a laboratory strain selected for resistance to azinphos-methyl [32]. Evidently, the last 20 years’ usage of chemical insecticides has modified the development of resistance [6]. An additional problem appeared in the mid-1990s with the development of cross-resistance due to the CM becoming resistant to several chemical groups of insecticides simultaneously [33].

Bosch et al. [34] determined the efficacy of new versus old insecticides against the CM in Spain. In their bioassays, they used 10 different active ingredients on twenty field populations of CM. Very high resistance ratios were detected for methoxyfenozide and lambda-cyhalothrin, while 50% of the populations were resistant or tolerant to thiacloprid. Tebufenozide showed very good efficacy in all the field trials. Even though CM showed resistance to chlorpyrifos-ethyl because of its widespread use, in this trial it was effective against CM populations. All other insecticides (indoxacarb, spinosad, chlorantraniliprole, emamectin, and spinetoram) provided high efficacy. These results showed that resistant CM populations in Spain can be controlled using new reduced-risk insecticides [34]. The newest and, at the same time, the first study of insecticide resistance and analysis about its resistance status in China showed insensitivity to chlorpyrifos-ethyl and carbaryl [35]. The first study of insecticide resistance in Greece showed reduced susceptibility to major groups of insecticides which were included in bioassays (azinphos-methyl, phosalone, deltamethrin, thiacloprid, fenoxycarb, tebufenozide, methoxyfenozide and diflubenzuron). But, also important, known target-site resistance mechanisms (kdr and modified AChE) were not detected [36].

Baculoviruses are insect pathogenic viruses that are widely used as biological control agents of insect pests in agriculture. One of the most important commercially used baculoviruses is the *Cydia pomonella* granulovirus (CpGV) [37]. For more than 30 years, commercial CpGV products have been successfully applied to control CM in organic and integrated fruit production. For all European CpGV products, the original Mexican isolate described by Tanada in 1964, CpGV-M, has been used [37]. According to Harison and Hoover [38], a granulovirus (GV) was identified from CM cadavers and found to be a type 2 GV that killed larvae in three to four days at higher concentrations. After promising field tests as a control measure in 1968 and 1977 [39,40], CpGV was developed into several control products in Europe and in North America. CpGV is used to control CM on over 100,000 ha of organic and conventional apple orchards in Europe [41,42]. Since 2005, resistance against the widely used isolate CpGV-M has been reported from different countries in Europe [41,43,44]. In a multination monitoring program, Schulze-Bopp and Jehle [45] identified that 70% of CM were resistant or partly resistant to CpGV across multiple orchards in Germany, Austria, Switzerland, Italy, and the Netherlands. The recent research by Sauer et al. [46] described autosomal and dominant inheritance of this resistance and demonstrated cross-resistance to different CpGV genome groups. The same authors report a CM field population with a new type of resistance, which appears to follow a highly complex inheritance in regards to different CpGV isolates [47]. In the European Union (EU) there are no strategic integrated pest management (IPM) programs that solve the current confusion surrounding CM control and resistance. There is a need for new control tools and a fresh approach to CM control and management in the EU.

## 3. Present Strategies in Codling Moth Suppression

### 3.1. Mechanical Control

Because of resistance development in CM populations, there is a need for alternatives to insecticides and CpGV. In recent studies, special attention is given to insect exclusion netting systems in apple production. The first netting system was designed in France in 2005 and in 2008 it was introduced in Italy. In both countries, a high level of efficacy of nets was observed against CM, especially for the ‘single-row’ system, which the authors recommend because it was more efficient and more durable than the ‘whole-orchard’ version. Also, this method enables a significant reduction in pesticide use without any major risks for apple production [48]. Pajač Živković et al. [49] tested the effectiveness of insect exclusion netting systems in preventing the attack of CM on apple fruits in Croatia. The authors showed a significant reduction in CM catches and also fruit injury compared to the non-netted control. This is consistent with similar studies in which nets significantly reduced the number of CM catches [50,51]. Modifying the orchard microclimate and reducing the interception of light using netting systems could have a negative consequence on the organoleptic quality of apple fruit according to Baiamonte et al. [52]. While the netting system prevents the entry of insect pests, it also serves as a barrier to beneficial insects (e.g., ladybugs, true bugs and syrphid flies) which could negatively affect natural pest control services. [49]. Alaphilippe et al. [48] recommend, due to the cost and constraints of netting, that this method be used in areas where CM is difficult to control.

### 3.2. Chemical Control

Chemical control of CM is still the main method used in integrated pome fruit production [53]. According to the Insecticide Resistance Action Committee (IRAC) [54] for CM control in most countries, there are 11 modes of action (MoA) available on the market depending on the country. For CM, some insecticides affect the nervous system, or pest growth and development. Acetylcholinesterase inhibitors (carbamates and organophosphates), sodium channel modulators (pyrethroids), nicotinic acetylcholine receptor agonists (neonicotinoids), nicotinic acetylcholine receptor agonists allosteric modulators (spinosyns), chloride channel activators (avermectins), voltage-dependent sodium channel blockers (oxadiazines) and ryanodine receptor modulators (diamides) all affect the pest’s nervous system; these insecticides are fast-acting [54]. Juvenile hormone mimics (phenoxyphenoxy-ethylcarbamate), chitin biosynthesis inhibitors—type 0 (benzonylureas) and ecdysone agonists (diacylhydrazines) all affect pest growth and development [54]. Insect development is controlled by juvenile hormones and ecdysone by directly perturbing cuticle formation/deposition or lipid biosynthesis. Such insect growth regulators are generally slow to moderately-slow acting [54].

From ca. the 1890s until today, insecticide groups and active substances used for CM suppression have been rapidly evolving. As can be seen from Table 1, chlorinated hydrocarbons, organophosphates, and carbamates were first used for the suppression of CM. Frequent applications of pyrethroids began in 1980 due to their lower toxicity to mammals and strong initial effect on insects. Although they are more environmentally friendly and can be applied in low doses per unit, area resistance has been observed. Microbial insecticides and insect growth regulators have been mostly used since the 1980s but after several years of application, resistance to them also occured. Since 2000 there have been a couple of new active compounds (i.e., chlorantraniliprole, spinetoram) that meet the requirements of integrated pest management (IPM) programs.

The classic model of CM suppression implies the intense application of aggressive chemical preparations, most commonly a wide spectrum of activity. Due to the altered biology of the CM (i.e., more generations/year) insecticides must be applied several times per season [57,58]. Some populations of CM have gained simultaneous resistance to several chemical subgroups of insecticides. In light of this and to delay resistance development, the rotation of compounds from different MoA groups ensures that repeated selection with compounds from any single MoA group is minimized. By rotation of insecticides across all available classes, selection pressure for the evolution of any type of resistance is minimized and the development of resistance will be delayed or prevented. The presence of kdr resistance renders pyrethroids less effective, whereas carbamates and organophosphates can still be used. In addition, the use of larvicides such as the organophosphate in conjunction with pyrethroids can support resistance management through rotation of MoA across different life stages. Effective long-term resistance management is important, but many factors have to be considered (including regional availability of insecticides). Currently, there are eight MoAs for CM control. In practice, it should not be difficult to implement rotation programs because there are enough active substances of insecticides in Europe that have mandated approval for CM. Alternatives to more persistent molecules are being developed [59,60]. For example, Bassi et al. [61] describe the development of a new compound, chlorantraniliprole, which belongs to a new class of selective insecticides. That makes chlorantraniliprole a valuable option for insecticide resistance management (IRM) strategies. Chlorantraniliprole is safe for key beneficial arthropods and honey bees, which renders it IPM compliant (i.e., excellent toxicity profile and use in low doses provide safety for consumers and agricultural workers). Nevertheless, there is a need for the improvement of alternative pest control methods, such as the application of microbial insecticides, mating disruptors or attract-and-kill methods. Production of high quality and healthy fruit that does not harm human health and the environment should continue to rely on an integrated production system where insecticide treatments must be applied responsibly and only when they are needed [62].

### 3.3. Biological Control

Biological control agents play a key role in most IPM strategies; these include entomopathogens, parasitoids and predators [63]. For augmentative biological control of CM, viruses such as granulovirus and entomopathogenic nematodes (EPNs) (*Steinernema carpocapsae*, *Steinernema feltiae*, *Heterohabditis* spp.) have been used as microbial agents [61].

The most widely used biopesticide is *Bacillus thuringiensis* (*Bt*) [64]. For controlling CM, *Bt* is very limited because of the improbability of ingesting a lethal dose of *Bt* toxin during feeding by neonate larvae [63]. On the other hand, granulovirus (GV) (Baculoviridae) is one of the most efficient and highly selective pathogens for suppression of CM. Its specificity for CM and safety to non-target organisms is documented by Lacey et al. [65]. It is one of the most virulent baculoviruses known. According to Laing and Jaques (1980) and Huber (1986), the LD_50_ for neonate larvae has been estimated at 1.2 to 17 granules/larva. The biggest disadvantage of CpGV is its sensitivity to solar radiation [66,67,68], and the need for frequent reapplication.

Parasitoids are insects whose larvae feed and develop within or on the bodies of other arthropods. Each parasitoid larva develops on a single individual and eventually kills that host [53]. Parasitoid wasps from the families Braconidae (*Ascogaster quadridentata* and *Microdes rufipes*), Ichneumonidae (*Mastrus ridibundus* and *Liotryphon caudatus*) and Trichogrammatidae (*Trichogramma* sp.) are the best known parasitoid species of CM. The parasitism of entomophagous wasps *M. ridibundus* and *A. quadradentata* has been successfully applied in CM control in some US states [63]. Species from Braconidae most commonly parasitize CM larvae, and Ichneumonidae parasitize CM larvae and adults and Trichogrammatidae parasitize eggs of Tortricidae moths. A reduction of 53–84% of CM was achieved by the experimental release of two *Trichogramma* species (*T. dendrolimi* and *T. embryophagum*) in apple orchards in Germany [53]. An additional benefit of the release of parasitoids is the simultaneous control of other pest species in apple orchards. The beneficial organisms alone can play an effective role in IPM but in general, the effect on CM control in economically productive orchards is considered insufficient [69].

For biological control, the most promising EPN species for suppression of CM are from the families Steinernematidae and Heterorhabditidae [70]. Species from both families are obligatorily associated with symbiotic bacteria (*Xenorhabdis* spp. and *Photorhabdis* spp., respectively) which are known for quickly killing its host insect. The most promising results for CM control have been with *Steinernema feltiae* and *Steinernema carpocapsae* [71]. Cocooned overwintering CM larva is the life stage most practical to control using EPNs. That life stage occurs between late summer and early spring in cryptic habitats, such as underneath loose pieces of bark or in pruning wounds on trees [71]. Eliminating cocooned larvae would protect fruit from damage in the following growing season [72]. The main obstacles for successful CM control with EPNs are low fall temperatures and desiccation of the infective juvenile stage of EPNs before they have penetrated the host’s cocoon.

Few studies exist on CM predators and biological antagonists. The largest group of CM predators are insects. Other important CM predators can be spiders, bats and birds [73,74,75]. In undisturbed habitats the eggs and neonate larvae of CM are most commonly preyed upon by small heteropteran insects, including: Anthocoridae, Miridae, *Phytocoris* sp., *Diaphnidia* sp., and *Deraeocoris* spp. Larger Carabidae and Dermaptera also play an important role [76]. The review of CM natural enemies and stages that are affected are summarized in Table 2.

Part of biological control is also ecological engineering, which includes the manipulation of farm habitats to be less favorable for arthropod pests and more attractive to beneficial insects [77]. To increase the activity of EPNs, ecological engineering encourages the use of environmental modification with mulches and irrigation [63]. Mulching is a strategy for conserving water and it is likely to become increasingly important for long-term sustainability in orchards [78]. In support of mulch, compared with bare ground, it may enhance CM control by providing cocooning sites for larvae, in a substrate that is easy to treat, maintains moisture and enhances nematode activity [72,79,80]. De Wall et al. [81] investigated the potential of using the EPN *Heterorhabditis zealandica* in combination with different mulch types (pine chips, wheat straw, pine wood shavings, blackwood and apple wood chips) to control diapausing CM. Their results showed that highest CM mortality was when they used pine wood shavings as mulch (88%) compared to pine chips, wheat straw, blackwood and apple wood chips (41–88%). Importantly, their research showed that humidity had to be maintained above 95% for at least 3 days to ensure nematode survival.

### 3.4. Population Genetic Monitoring

Analysis of population genetic structure is a key aspect in understanding insect pest population dynamics in agriculture [82]. The development of effective pest management strategies relies on a multidisciplinary approach [83] and one component of this is knowledge of the population genetics of the pest. Genetic structure and patterns of dispersal at the local and landscape scale are important for establishing a control strategy for insect pests [84]. Understanding the population genetics of CM invasions enables identification of the geographic origin, number of introduction events and the spread of the infestation [85]. According to Keil et al. [86] CM populations are composed of mobile and sedentary genotypes and this has direct consequences for the local observable population dynamics of the species as well as the implementation of new behavior-based pest management measures (e.g., mating disruption, attract-and-kill and SIT technique) [87]. The first attempt to elucidate the population genetic structure of CM on a global geographic scale (i.e., inter-continental) using allozymes was conducted by Pashley and Bush [88]. These authors showed that CM populations were not differentiated among countries investigated (F_ST_: 0.05). Following this, Bues and Toubon [89] used the same approach to study populations in Switzerland and France. More recently, Timm et al. [90] and Thaler et al. [7] used amplified fragment length polymorphism (AFLP) markers to study the molecular phylogeny and genetic structure of CM where they found large differences among these populations (F_ST_: 0.70). More recently, co-dominant microsatellite markers from CM were developed by Zhou et al. [91] who characterized 17 loci. An additional 24 microsatellite loci were characterized by Frank et al. [92], with these loci most frequently used in population genetic studies worldwide [6,15,82,84,93].

Franck et al. [6] used those markers to investigate the genetic structure of CM populations from 27 orchards from three continents (Europe, Asia and South America) to determine the dynamics of CM meta-populations and the impact that human activities had on these dynamics. Franck et al. [6] showed that populations of CM are structured by geographic distance on the intercontinental level. However, analyses of CM populations from treated and untreated orchards in Europe and South America (France and Chile) did not show significant genetic differentiation by country, but rather a pattern of minor influence of insecticide treatments on allelic richness. A similar comparison of CM genetic structure from treated versus untreated populations using microsatellite markers (following Franck et al. [6]) was conducted in Croatia [15]. Even though differences in genetic structure among populations were low and not statistically significant, untreated populations of CM had the highest average number of alleles and the largest number of unique alleles compared to treated populations. Overall, the study’s findings suggested a possible reduction of allelic richness in treated populations due to the frequent application of insecticides. The authors have questioned whether these genetic changes may relate to the increase in reproductive abilities of CM and a change in its overall biology in Croatia [15].

Frank and Timm [82] also used microsatellite markers to study CM genetic structure and gene flow from organic versus treated apple orchards. They found low genetic variation between populations but significant partitioning of genetic variation within individuals. Chen and Dorn [93] used nine microsatellite markers to investigate genetic differentiation and the amount of gene flow between populations from orchards in Switzerland and laboratory populations. They noted significant genetic differentiation among populations from apple, apricot and walnut orchards and also between populations collected from orchards that were less than 10 km apart. These results are consistent with Timm et al. [90] and Thaler et al. [7] and provide significant evidence for CM population differentiation at small spatial scales, even within the same bio-region. Fuentes-Contreras et al. [94] found significant but weak genetic differentiation between populations across time and space comparisons. These authors found no significant correlation (r: −0.03; *p*: 0.56) between genetic distance and geographic distance of the studied populations and the lack of structure at a local scale with frequent adult movement between treated and untreated orchards. Also, their data highlights the importance of developing area-wide management programs for successful CM control. Men et al. [95] used eight microsatellite loci to infer the characteristics of genetic diversity and genetic structure of 12 CM populations collected from the main distribution regions (Xinjiang, Gansu and Heilongjiang Provinces) in China and compared them with one German and one Swiss population.

They found ascertained loss of genetic diversity and important structuring related to distribution, however no important correlation between genetic distance and geographic distance among populations (F_ST_: 0.22091) was found. Voudouris et al. [96] used 11 microsatellite loci to analyze nine CM populations from Greece and six from France for comparison. Results from Bayesian clustering and genetic distance analyses separated CM populations in two genetic clusters. In agreement with previous published studies F_ST_ values showed low genetic differentiation among populations (Greek populations F_ST_: 0.009 and F_ST_: 0.0150 French populations).

Dispersal of fertilized females is important because it directly affects the effectiveness of pest control programs. Margaritopoulos et al. [97] used the mark-release-recapture (MRR) method on male and female individuals from two laboratory and one wild CM populations. Kinship analysis was made on 303 genotyped individuals (11 microsatellite loci) from two contiguous apple orchards to see the dispersal patterns in the Greek CM populations. The collected data confirm the view of the sedentary nature of CM and indicate that genotypes able to migrate at long distances are not present in the studied area. The information obtained could be fundamental for determining the dynamics and genetics of the pest populations and for developing efficient management programs. Results about the dispersal pattern of codling moths might have practical applications in mating disruption or mass trapping pest control programs.

### 3.5. Area-Wide Integrated Pest Management

The 5-year CAMP (CM Area-Wide Management Program) was the first of the area-wide programs initiated by the US Department of Agriculture [98]. Demonstration of this was initiated in 1995 in a multi-institutional program created through the collaboration of university and government researchers in Washington, Oregon and California. The goal of this program was to implement, assess, research and educate industry users about promising new IPM technologies. CAMP was highly successful in fueling the rapid adoption of a new paradigm in orchard pest management that resulted in significant reduction in fruit injury using nearly 80% less broad-spectrum insecticides [95].

IPM is based on environmentally and toxicological acceptable treatments. Using pheromones, attract-and-kill methods and mating disruption results in a promising way of controlling CM. According to Witzgall et al. [99], orchard treatments with up to 100 g of synthetic pheromone per hectare effectively control CM populations over the entire growing season. The disadvantage of these techniques is that females are not affected [100].

After Roelofs et al. [101] identified the main pheromone components for CM attraction (i.e., E8, E10-dodecadienol (codlemone)), pheromone traps started to be a useful tool for insect detection and monitoring and later for its suppression. Mating disruption is based on tactics to employ synthetic sex pheromones that interfere with the ability of males in finding female moths and as a control strategy it shows considerable promise. Currently, it is used to suppress CM populations in over 160,000 ha of apple and pear orchards worldwide [99]. The first commercially available pheromone dispenser for control of CM was Isomate-C^®^, which became available in the USA in 1991 [55]. Monitoring of CM in orchards treated with sex pheromone mating disruption (MD) has become widely adopted and is very important for its effective management [99]. Traps used for monitoring are baited with the sex pheromone (*E*,*E*)-8,10-dodecadien-1-ol (codlemone) that attracts males [102] and ethyl (*E*,*Z*)-2,4-decadieonate, a pear-derived kairomone, to attract both sexes of CM [103]. The combination of pear ester with codlemone (PH-PE) in a lure is effective for monitoring both sexes of codling moth in sex pheromone-treated orchards. Monitoring females, instead of only male CM, has certain benefits, like egg density and timing of egg hatch. A number of studies have used pear ester’s attractiveness for both male and female CM to develop alternative approaches to further enhance the catch of female moths [104,105,106]. Using pear ester with acetic acid (AA) can increase moth catches, especially of females [107]. The co-emission of acetic acid improves the capture performance of pear ester in clear traps to levels equivalent to the PH-PE lure when used in orchards treated with sex pheromone dispensers [108]. The effectiveness of this mating disruption as a technique depends on numerous factors (shape, size, isolation and environment of orchards) as well as the starting density of the CM population itself. In order for mating disruption to be successful there is a need for low CM population levels and a reliable monitoring system [109]. Mating disruption for CM began in the US in 1995 in large contiguous apple blocks (400 ha) and small private orchards [110]. According to Witzgall et al. [99] and Casado et al. [111], Europe also does not lag far behind in its application of this technique. In Croatia, this method is not widely used, although the first field trials in 1999 and 2000 [112] were promising and did reduce the number of insecticides being used during those growing seasons. Barić and Pajač Živković [113] showed that the highest protection efficacy was achieved with 92.65% control in the standard part of the orchard, and the efficacy of mating disruption was 67.65% and 73.53%. Although the authors concluded that this method of control was not economically justifiable given the high cost (approx. 150 €/ha) of protection and first-class fruit losses. However, their results also confirmed that the mating disruption method must be combined with the application of two insecticide treatments to increase the efficacy and profitability of apple production. Miller and Gut [114] agree that pest control by mating disruption is an important and growing industry. This combined control of CM is more ecologically oriented and also meets the toxicological minimum requirements of the food suppliers and the food retail chain. They propose some key economic and policy questions that will require the collective efforts of scientists and society as a whole if the benefits of mating disruption are to be maximized. There is still a lot of work to be done to optimize the role of mating disruption as one of the components of modern integrated pest management.

Mass trapping, as one of the first mating control strategies, can significantly reduce CM damage levels. However, several intensive field studies have shown that it is not effective enough for CM control because of the low damage thresholds (no more than 1–2% of the crop) required in commercial apple growing. Since adequate control cannot be achieved by using only mass trapping, there is a need for combining it with other control measures [115]. Another problem is the cost and practical difficulties of deploying sufficient trapping stations. If droplets containing sex pheromones and a fast-acting insecticide are used instead of traps [116], then the costs can be substantially reduced. The potential strength of the approach is that males have been removed from the system, stopping their ability to find a mate.

The attract-and-kill method, in its technically simplest form is the attractant applied as a ‘tank-mix’ with an insecticide. This method uses the same attractants as mass trapping but in an envelope impregnated with an insecticide on the outside. This technology has shown efficacy in the control of several important lepidopteran pests including pink bollworm, *Pectinophora gossypiella* (Saunders), light brown apple moth, *Epiphyas postvittana* (Walker), and CM [117]. In both systems, mass trapping and attract-and-kill, chemicals are utilized only when the population increases considerably [118].

For AW-IPM the integration of sterile insects is a very effective and environmentally friendly control tactic that can be combined with other control practices and offers great potential [119,120]. Sterile insect technique (SIT) is non-destructive to the environment, does not affect non-target organisms, and can easily be integrated with other biological control methods such as parasitoids, predators and pathogens [121]. The technique has gained traction in the last few decades [122,123]. SIT is an autocidal pest control technique that controls pests with a form of birth control [121]. The target pest species is mass-reared, sterilized through the use of gamma radiation and then released in the target area in high numbers. After release, sterile males will locate and mate with wild females and transfer the infertile sperm thus reducing the wild population. Another method of sterilization is genetic manipulation or sexing strains, where lethal mutations are incorporated into sperm [121]. The SIT, together with mating disruption, granulosis virus and EPNs, are the options that offer great potential as cost-effective additions to accessible management techniques for AW-IPM approaches.

In Table 3, a review of changes in the suppression of CM through the last two decades and factors that affect the current scenario in comparison to the year 2000 is shown. Reduction of chemical control measures due to EU regulations and food chain pressures, increased adoption of semiochemicals for mating disruption, and microbial insecticides contributed to the suppression of CM. Improved investigation tools for resistance detection and confirmatory assays have contributed to the decrease of field resistance issues and better knowledge of resistance.

## 4. Resistance Management Strategies

The most effective strategy to combat insecticide resistance is to do everything possible to prevent it from occurring in the first place. To this end, crop specialists recommend insect resistance management (IRM) programs as one part of a larger (IPM) approach covering three basic components: monitoring pest complexes in the field for changes in population density, focusing on economic injury levels and integrating multiple control strategies. IRM is the scientific approach of managing pests long term and preventing or delaying pest evolution towards pesticide resistance and minimizing the negative impacts of resistance on agriculture [124]. The basic strategy for IRM is to incorporate as many different control strategies as possible for particular pests including the use of synthetic insecticides, biological insecticides, beneficial insects (predators/parasitoids), cultural practices, transgenic plants (where allowed), crop rotation, pest-resistant crop varieties, and chemical attractants or deterrents. The establishment of an anti-resistance program in perennial crops is slightly more difficult than in arable crops where crop rotation is possible. If non-chemical methods provide satisfactory pest control, preference should be given to them over chemical methods. Key insect pests of apple and grape such as CM and grapevine moths are effectively controlled via mating disruption. In Switzerland, mating disruption is in use in 50% of the apple orchards and 60% of vineyards, and this has enabled a reduction of synthetic pesticide use by two thirds [125].

Insecticides, if necessary, must be selected with care and their impact on future pest populations considered. Broad-spectrum insecticides should always be avoided when a more specific insecticide will suffice. Even cultural practices, such as irrigation for destroying overwintering stages (e.g., cotton bollworm, *Helicoverpa armigera*) of pests can play a role in managing resistance [126]. When insecticide is applied it should be timed correctly and for the best efficacy, it should target the most vulnerable life stage of the insect pest. It is important to mix and apply insecticides carefully. With the increasing problem of resistance, there is no space for error in terms of insecticide dose, timing, coverage, etc.

Reducing doses, application frequency, and resorting to the partial application of pesticides contribute to the IPM goal of reducing or minimizing risks to human health and the environment. Regular monitoring for insecticide resistance is essential to react proactively to prevent insecticide resistance from compromising control [127].

Before applying any CM control action, it is necessary to monitor CM occurrence and early infestation of apples. Pheromone traps are used in orchards to determine the present amount of adult male moths. For estimating the potential infestation risk of the second generation, it is recommended to examine 1000 young apples in June for damage or the presence of CM [128]. Spray thresholds are also based on the number of moths in the pheromone traps or on infestation rates detected in the harvest of the current or last season. For apples, the economic threshold for the CM is 1% of infested fruit [55].

Figure 1 shows recommendations for effective CM control and resistance management based on current knowledge: I. to monitor; II. application of ecotoxicological favorable protection measures like mating disruption (when CM population levels are low); III. application of chemical control measurements (if necessary); and IV. control of overwintering stages by applying biological agents (e.g., CpGV, nematodes) to reduce the late summer and fall CM population in order to minimize the population in the following growing season. It is an effective example of how resistance management should work in orchards (Figure 1).

## 5. Perspectives in Codling Moth Resistance Detection

Reliable data on resistance are essential to successful resistance management. Bioassay is a method used for evaluating the status of resistance in insect populations. Effective resistance management relies on sound information about the extent and intensity of resistance problems [128]. There are several different bioassay methods to monitor for CM resistance, such as diagnosing metabolic resistance using differential enzymatic activity between life-stages within the same population. The analysis of the enzymatic activity (MFO, GST, EST) in a CM population is a key element for resistance evaluation [54]. In the last decade, large-scale monitoring for field resistance mostly relied on topical application to diapausing codling moth larvae. Recent studies have confirmed their validity for IGRs but questioned their reliability for the prediction of field resistance with some neurotoxic insecticides [54]. Bioassay of the target-stage includes resistance monitoring done on the target instar. For larvicidal products, ingestion bioassays on neonate larvae (F1 or F2 of the feral population), IRAC method no. 017, normally provide a more reliable indication of the field situation than topical application to diapausing larvae [54].

So far, the only approved method for CM sensitivity monitoring is IRAC method 017 [54]. This method is specifically recommended by the IRAC Diamide Working Group for evaluating the susceptibility status of diamide insecticides (IRAC MoA 28). Also, it is suitable for the following insecticide classes (IRAC MoA class): organophosphate (1B), pyrethroid (3A), neonicotinoids (4A), spinosyn (5), avermectin (6), juvenile hormone mimics (7A), fenoxycarb (7B), benzyl urea (15), diacylhydrazine (18), indoxacarb (22A), metaflumizone (22B), and pyridalyl (un) [54]. According to this method, the first step is to collect a representative sample of insects from a field. These may be larvae, pupae or adults for rearing to the appropriate stage from which an F1 population for testing can be reared. A minimum of 100 larvae or diapausing pupae should be collected for each population to be tested, to establish a breeding colony of at least 50 adults. When we have enough CM larvae for the bioassay, the second step is to prepare an accurate dilution of the test compound from the identified commercial product. Six evenly spaced rates allowing a clear dose-response are recommended [54]. For this method, a single neonate (less than 24 h old) of CM larvae should be used. In the case of diamide insecticides, organophosphates (1B), pyrethroids (3A), neonicotinoids (4A), spinosyns (5), avermectins (6), indoxacarb (22A), metaflumizone (22B) and pyridalyl (un), a final assessment of larval mortalities (dead and live) is made after 96 h. For juvenile hormone mimics (7A), fenoxycarb (7B), benzyl urea (15) and diacylhydrazine (18), a 120-h assessment period should be used. Also, larvae should go through full molt before the mortality assessment [54]. The number of dead larvae and moribund larvae (seriously affected larvae which are unable to make coordinated movement and cannot return to an upright position when turned upon their backs with a seeking pin or fine-pointed forceps) are to be summed and considered as dead. Results should be expressed as percentage mortalities, correcting for “untreated” (control) mortalities using Abbott’s formula [54].

Through innovation it is possible to establish reliable strategies for detecting resistant CM populations. Of most importance is the timely detection of resistant populations in order to suppress them and prevent further spread of resistance. For this purpose, exploration of existing tools, though with novel use as monitoring tools, is warranted (i.e., geometric morphometrics and population genomics).

Geometric morphometrics (GM) offers a powerful method for studying intraspecific variation or ecotypes and it has been shown to be a useful bio-monitoring tool [129]. It is known that metric properties (wing shape and size) are the first morphological characters to change as influenced by environmental and genetic factors [130,131]. This therefore makes them an ideal technique to detect and monitor population variation and resistant variants in the field [132,133]. Furthermore, the use of GM generates important new data on basic insect biology and ecology.

Recently, wing or body shape and size has been used as a population bio-marker to detect: differences between susceptible and resistant variants [134]; population changes related to invasion [135]; and morphological differences in resistant versus non-resistant populations and rotation versus *Bt*- resistant strains of western corn rootworm [136]. GM was tested as an existing method, though novel in its application, for morphological differences in field-insect pest populations versus laboratory populations and integrated versus ecological populations in Croatia. That is, Pajač Živković et al. [137] revealed two noticeable wing shape morphotypes in *Drosophila suzukii* (i.e., vein configuration) between grape and strawberry crops. Different IPM practices in agro-ecosystems generate different degrees of disturbance in insect communities, as shown by Benitez et al. [138] where shape variation and fluctuating asymmetry levels were estimated by applying GM methods to the beetle *Pterostichus melas melas*.

Specifically, for CM, Khaghaninia et al. [139] used GM methods as tools to show significant differences in CM fore and hindwings as a function of season (overwintered vs. summer), geographic location and sex. Also, Pajač Živković et al. [140] investigated the relationship between different pest management types and CM morphology using GM. The authors detected population changes related to different types of apple production. The aforementioned publications provide compelling evidence for the use of GM as a population bio-marker when applied to CM and other insect pest monitoring.

Recent enhancements with the speed, cost and accuracy of next generation sequencing are revolutionizing the discovery of single nucleotide polymorphisms (SNPs) and field of population genomics. SNPs are increasingly being employed as the marker of choice in the molecular ecology toolkit in non-model organisms. SNPs are attractive markers for many reasons [141,142], including: the availability of high numbers of annotated markers; low-scoring error rates; relative ease of calibration among laboratories compared to length-based markers; and the associated ability to assemble combined temporal and spatial data sets from multiple laboratories.

SNPs are single base substitutions found at a single genomic locus. Although they have lower allelic diversity and provide less statistical power to discriminate unique genotypes, they have a denser and uniform distribution within genomes which makes them very useful for population genetic studies. In recent times, SNPs have become an affordable and readily accessible means of generating a lot of data quickly for non-model species [143]. Genotyping of SNPs has potentially far-reaching applications in insect population genomics. SNP detection has facilitated association mapping studies in many insect species including: *Drosophila melanogaster* [144], *D. v. virgifera* [145], *Aedes aegypti* [146], *Glossina fuscipes* [147], *Diatraea saccharalis* [148], *Phaulacridium vittatum* [149] and other insects in which specific nucleotides are statistically associated with complex phenotypic traits. Detailed genomic data could provide an answer about genetically conditioned resistance development in insects. By combining genetic and GM population monitoring, it may be possible to identify the addition or deletion of alleles and different haplotypes, and the genetic and morphometric patterns which have developed under the selective pressure of control.

## 6. Conclusions

CM is the most harmful insect species of the Tortricidae family that causes economic damage to apple production worldwide. The suppression of this pest in the past relied on intensive insecticide application(s) which ultimately led to the development of resistance and caused a decrease in population of beneficial species which were once the only natural regulators of pest populations in apple farming. One of the basic goals of integrated production is growing high quality and healthy fruits that contain minimal residues of pesticides; such production is safer for human health and the environment. To achieve this goal, environmentally friendly area-wide IPM strategies must be established. This involves the use of pheromones and kairomones (attract-and-kill methods and mating disruption) and sterile males (SIT technique) which combined with the use of natural enemies (mainly viruses and nematodes) serve as good alternatives to chemicals. Also, recent advancements in the use of mechanical protection measures against CM (insect-proof nets) have shown very promising results in field trials. All available control measures against CM should be used in combination and there should be an informed and systematic strategy for their use. Effective IRM strategies should involve all available tools for pest control (e.g., natural enemies, biotechnical tools, alternative insecticides) and make a concerted effort to trial and use existing technologies, though with novel applications (e.g., GM for monitoring population phenotypic changes and SNPs for monitoring population genetic changes) for their monitoring, therefore fulfilling the best practice resistance management strategy discussed here.

## Figures and Tables

**Figure 1 insects-11-00038-f001:**
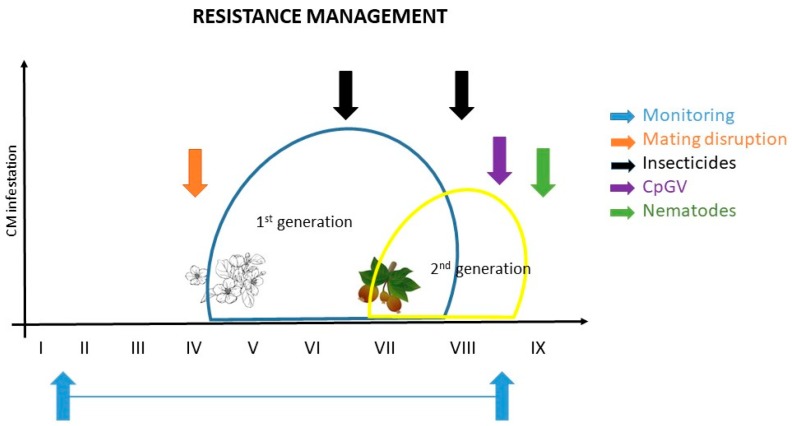
Example of resistance management for codling moth; the ideal control is a combination of different measures (modified by Martina Kadoić Balaško).

**Table 1 insects-11-00038-t001:** Review of registered insecticides to suppress codling moth from 1890–current [54,55] and time of resistance development according to the Arthropod Pesticide Resistance Database [56].

Insecticide Group	MoA [54]	Insecticide/Active Substance	Use Period (Approximate)	Resistance Development (Year of First Report/Region)
Inorganic/others		Arsenate	1890s–1950s	1928/USA
		Nicotine	1960s	
Chlorinated hydrocarbons		DDT	Mid 1940s–1970s	1955/USA
		Thiodan/Endosulfan	1960s–1970s	1965/Syria
Organophosphates	1B	Diazinon	1950s–2000s	
		Phosalone	1960s–2000s	
		Azinphosmethyl	1950s–present	1991/USA
		Chlorpyrifos-ethyl	1960s–present	2011/France, Spain
		Chlorpyrifos-methyl	1960s–present	
		Methidation	1950s–1990s	
		Phosmet	1970s–present	1999/USA
		Mevinphos	Mid 1950s–mid 1990s	
		Methomyl	1970s–1990s	
		Oxamyl	Mid 1980s–1990s	
		Formetante hydrochloride	1970s–1990s	
Charbamates	1A	Carbaryl	1970s–present	2012/Spain
Pyrethroids	3A	Fenvalerate/Esfenvalerate	1970s–present	
		Permethrin	1970s–present	
		Bifenthrin	1980s–present	
		Deltametrin	1970s–present	2001/China
		Flucythrinate	1980s–present	
		Lambda-cyhalotrin	1980s–present	2008/USA
		Gama-cyhalotrin	1980s–present	
		Tau-fluvalinate	1980s–present	
Microbial insecticides		Bacillus thuringiensis sub sp. kurstaki	1980s–present	
		Codling moth granulovirus (CpGV)	1980s–present	2007/Germany
Naturalites	5	Spinosad	1990s–present	
Insect growth regulators	15	Benzonylureas (diflubenzuron, hexaflumuron, flufenoxuron, triflumuron, lufenuron, teflubenzuron)	1970s–present	diflubenzuron/1988/USA triflumuron/1995/France teflubenzuron/1995/France flufenoxuron/2011/Spain
	7B	Fenoxycarb	1980s–present	2007/Czechoslovakia
	18	Tebufenozide	1990s–present	1995/France
		Methoxyfenozide	1990s–present	2008/USA
	7B	Pyriproxyfen	2000–present	
Nicotinoids	4A	Acetamiprid	1990s–present	2010/USA
		Thiacloprid	2001–present	2011/Spain
		Thiamethoxam	2001–present	
Avermectins	6	Emamectin benzoate	2000–present	
Anthranilic diamide insecticides	28	Chlorantraniliprole	2007–present	
Spinosyns	5	Spinetoram	2011–present	

**Table 2 insects-11-00038-t002:** Review of codling moth natural enemies and life stage attacked [63].

Natural Enemies	Organism/Family	Family/Species	CM Life Stage Attacked
Entomopathogenic organisms	Virus	Granulovirus (CpGV)	Neonate larvae
	Bacteria	*Bacillus thurigiensis*	Neonate larvae
	Fungi	*Beauveria bassiana*	Cocooned overwintering larvae
	Nematodes	Steinernematidae	Cocooned overwintering larvae
		Heterorhabditidae	
Predators	Anthocoridae	*Orius insidiosus*	Eggs and neonate larvae
		*Anthocoris musculus*	
	Miridae	*Hyaliodes harti*	
		*Phytocoris* sp.	
		*Diaphnidia* sp.	
		*Blepharidopterus angulatus*	
		*Deraeocoris* spp.	
	Reduviidae		Mature larvae
	Nabidae		
	Carabidae, Trogossitidae, Malachiidae, Staphylinidae, Cleridae, Cantharidae, Elateridae		Cocooned larvae
	Formicidae		Mature larvae
	Phlaeothripidae	*Haplothrips faurei*	Eggs
		*Leptothrips mali*	
	Dermaptera	*Forficula auricularia*	
Parasitoids	Braconidae	*Ascogaster quadridentata*	Larvae
		*Microdes rufipes*	
	Ichneumonidae	*Mastrus ridibundus*	Larvae and adults
		*Liotryphon caudatus*	
		*Pimpla turionellae*	Pupae
	Trichogrammatidae	*Trichogramma* sp.	Eggs

**Table 3 insects-11-00038-t003:** Changes in codling moth control from 2000 until now (modified according to IRAC [54]).

	2000	2012	2017
No. of MoA available for codling moth control *	8	10	11
No. of individual insecticides available **	High	Decreasing	Fewer
Use of semiochemicals (mating disruption)	Minor	Moderate	Increasing
Microbial insecticides	Minor	Moderate	Moderate
Biological control	Minor	Minor	Minor
Regulatory pressure	Low	High	Decreasing
Food chain pressure	Low	High	Decreasing
Field resistance issues **/***	Moderate	Decreasing	Low
Resistance knowledge and investigation tools	Moderate	Increasing	High

* According to IRAC Mode of Action (MoA) classification, four MoA were introduced from 1997–2000, and two during 2007–2010. ** Number of individual insecticides available is decreasing every year. The criteria introduced in the revision of EU Directive 91/414 may concern a significant number of available insecticides, with an impact on sustainable control options. *** Dependent on the implementation of the other factors. The assumption is that sustainable insecticide use will continue to be possible and implemented. In this respect, increased use of non-chemical tools will play a key role.

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
