# Peer review of "Pest Management Challenges and Control Practices in Codling Moth: A Review"

_insects, 2020, doi:10.3390/insects11010038_

Round 1

Reviewer 1 Report

It has been increasing number of papers cover pest controls using reproductive manipulation using genetic modificationS. This paper is the same line of work focusing on a unique species. It is important to have more cases to be introduced. Thus I recommend the publication. however, the paper would be more complete if the actual method is explained in detail.

Author Response

Dear reviewer,

I would like to thank you for your time and effort in reading our article. Also, for your comments and suggestions.

Regarding your suggestion to explain the actual method in detail, we agree with you but we have one more article in progress about codling moth. We are just waiting for the results from SNPs analyses. In that work, our plan is to focus more on describing methodology.

I hope this answer satisfies you.

Sincerely,

Martina Kadoić Balaško

Reviewer 2 Report

Dear Authors,

I believe that your abstract overstates the depth of this paper. The sentence : "this review summarizes the information about the origin and biology of the codling moth, describes the mechanisms of resistance in this pest, and provides an overview of current research of resistant pest populations and genetic research both in Europe and globally." implies that you have collected all available know knowledge on this subject. That is clearly not the case. I believe that you have only scratched the surface, and do not fully understand the subjects of mating disruption, SIT, and efforts at genetic sterilization. I would encourage you to read: Mating Disruption for the 21st Century: Matching Technology with Mechanism, J.R. Miller, L J. Gut. 2015, among others. That being said, I would like to see papers come out of places other than EU and USA. If the claims made in the above quoted sentence are scaled back, and the indicated corrections made, I believe it could be published. 

Author Response

Dear reviewer,

I would like to thank you for your time and effort in reading our article. Also, for your comments, suggestions and especially for recommending Miller and Gut work.

Research about codling moth is a really wide area. Of course, we couldn’t have collected all available knowledge and could not cover all areas equally. We hope that the changes we made will be at least a little satisfying for you.

Please see the attachment for point-by-point responses.

Sincerely,

Authors

Reviewer 3 Report

Overall, the article reads very well.

I attached a pdf file with my comments as annotations inserted in the file.

Line 336: Add a short review about kairomone-based lures including pear ester (DA) and acetic acid (AA) for monitoring CM and potential for mass trapping both CM female and male moths.

line 390: comment on any literature about transgenic apples - Is there any literature noting potential of using transgenic host plant resistance in apple against CM?

line 420: There is some potential for using late summer/fall applications of CpGV to help reduce the CM population entering overwintering. You could modify figure 1 - add a different colored arrow for CpGV to note timing of a CpGV application and add CpGV to legend.

Author Response

Dear reviewer,

I would like to thank you for your time and effort in reading our article. Also, for your comments and suggestions.

Please see the attachment for the point-by-point responses.

Sincerely,

Martina Kadoić Balaško

Round 2

Reviewer 2 Report

Thank you for including suggested corrections. Figure one still requires a y-axis label. Please correct this final item.

Thank you.

Author Response

Dear reviewer,

We added the y-axis label as suggested.

We wish to thank you one more time for your comments and suggestions.

Sincerely,

Authors